# Leukocyte Nuclear Morphology Alterations in Dilated Cardiomyopathy Caused by a Lamin AC Truncating Mutation (*LMNA*/Ser431*) Are Modified by the Presence of a LAP2 Missense Polymorphism (*TMPO*/Arg690Cys)

**DOI:** 10.3390/ijms232113626

**Published:** 2022-11-07

**Authors:** Antonia González-Garrido, Sandra Rosas-Madrigal, Arturo Rojo-Domínguez, Jaime Arellanes-Robledo, Enrique López-Mora, Alessandra Carnevale, Leticia Arregui, Rigoberto Rosendo-Gutiérrez, Sandra Romero-Hidalgo, María Teresa Villarreal-Molina

**Affiliations:** 1Laboratorio de Enfermedades Mendelianas, Instituto Nacional de Medicina Genómica, Mexico City 14610, Mexico; 2Departamento de Ciencias Naturales, Universidad Autónoma Metropolitana, Unidad Cuajimalpa, Mexico City 05348, Mexico; 3Laboratorio de Enfermedades Hepáticas, Instituto Nacional de Medicina Genómica, Mexico City 14610, Mexico; 4Dirección de Cátedras, Consejo Nacional de Ciencia y Tecnología (CONACyT), Mexico City 03940, Mexico; 5Clínica de Insuficiencia Cardiaca, Instituto Nacional de Cardiología “Ignacio Chávez”, Mexico City 14080, Mexico; 6Departmento de Genómica Computacional, Instituto Nacional de Medicina Genómica, Mexico City 14610, Mexico; 7Laboratorio de Genómica Cardiovascular, Instituto Nacional de Medicina Genómica, Mexico City 14610, Mexico

**Keywords:** dilated cardiomyopathy, lamin AC, LAP2α, *TMPO* Arg690Cys, nuclear morphology, genetic modifier

## Abstract

The clinical phenotype of *LMNA*-associated dilated cardiomyopathy (DCM) varies even among individuals who share the same mutation. *LMNA* encodes lamin AC, which interacts with the lamin-associated protein 2 alpha (LAP2α) encoded by the *TMPO* gene. The LAP2α/Arg690Cys polymorphism is frequent in Latin America and was previously found to disrupt LAP2α-Lamin AC interactions in vitro. We identified a DCM patient heterozygous for both a lamin AC truncating mutation (Ser431*) and the LAP2α/Arg690Cys polymorphism. We performed protein modeling and docking experiments, and used confocal microscopy to compare leukocyte nuclear morphology among family members with different genotype combinations (wild type, LAP2α Arg690Cys heterozygous, lamin AC/Ser431* heterozygous, and LAP2α Arg690Cys/lamin AC Ser431* double heterozygous). Protein modeling predicted that 690Cys destabilizes the LAP2α homodimer and impairs lamin AC-LAP2α docking. Lamin AC-deficient nuclei (Ser431* heterozygous) showed characteristic blebs and invaginations, significantly decreased nuclear area, and increased elongation, while LAP2α/Arg690Cys heterozygous nuclei showed a lower perimeter and higher circularity than wild-type nuclei. LAP2α Arg690Cys apparently attenuated the effect of *LMNA* Ser431* on the nuclear area and fully compensated for its effect on nuclear circularity. Altogether, the data suggest that LAP2α/Arg690Cys may be one of the many factors contributing to phenotype variation of *LMNA*-associated DCM.

## 1. Introduction

Dilated cardiomyopathy (DCM) is characterized by left or biventricular dilatation and systolic dysfunction in the absence of underlying disease causing cardiac muscle overload and is the most frequent indication for cardiac transplantation [1]. The major clinical manifestation is heart failure, which is frequently associated with arrhythmia and sudden cardiac death [2]. Between 30 and 50% of DCM cases are of genetic etiology, most frequently with an autosomal dominant inheritance pattern, and more than 50 genes have been associated with DCM. A well-recognized clinical feature of inherited cardiomyopathy is variable phenotypic expression, with age-dependent penetrance and variable clinical presentations, even among patients sharing identical primary mutations [1,3]. This suggests that the final phenotype is influenced by modifier genes and environmental factors. Among the modifier genes involved in this variability, both protein-coding variants and variants located within enhancers have been described as altering the phenotypic expression of cardiomyopathy-causing mutations [4,5,6].

*LMNA* mutations cause 6 to 8% of familial DCM cases and are associated with poor prognosis, and accelerated and aggressive progression of heart failure and ventricular arrhythmias, which may result in sudden cardiac death at early ages [2,7]. The development of the phenotype occurs between ages 20–39 years in two-thirds of cases, with complete penetrance at age 60 years [8]. The *LMNA* gene encodes lamin A (661 amino acids) and lamin C (572 amino acids), classified as type V intermediate filament nuclear envelope proteins. Several hundred *LMNA* gene mutations have been linked to at least four different disease groups, collectively called laminopathies. These include striated muscle diseases, such as Emery–Dreifuss muscular dystrophia, lipodystrophies, metabolic diseases such as Dunnigan-type familial partial lipodystrophy, peripheral neuropathies such as Charcot–Marie–Tooth Type 2B1, premature aging syndromes such as Hutchinson–Gilford progeria, and DCM (OMIM 150330). The hallmark of laminopathies is the presence of nuclear abnormalities in different cell types, including blebs, herniations, and honeycomb-structures [9]. Leukocytes are one of several cell types that show nuclear morphology alterations in DCM patients with *LMNA* mutations [10,11]. However, the molecular mechanisms are not fully understood and are most likely explained by a combination of functional effects that are cell/tissue-specific [12,13,14,15,16,17,18].

Lamin AC proteins polymerize into complex filamentous meshworks associated with the internal nuclear membrane, and in less structured forms throughout the nucleoplasm, apparently playing a central role in chromatin organization and gene expression [19]. Moreover, nuclear envelope lamin AC is responsible for nuclear membrane assembly, structure, and shape, and provides protection from deformations caused by mechanical stress. It is involved in a broad range of functions, including mechano-transduction, DNA repair, signaling, and cell cycle regulation during development [20,21,22,23,24,25,26]. Additionally, DCM-associated *LMNA* mutations have been found to alter the nuclear lamin structure, protein interactions, and gene expression [27]. In a recent review, Crasto et al. stated that despite the plethora of studies describing the involvement of lamins in many nuclear and cellular processes, none of these can fully justify the DCM phenotype [28].

Lamins A and C interact with other lamins, chromatin, and regulatory proteins such as the lamin-associated polypeptide 2 (LAP2), apparently in a tissue-specific manner [29]. LAP2 is encoded by the *TMPO* gene and has 3 different isoforms (α, β and Ϫ). The LAP2α isoform binds with nucleoplasmic lamin AC during interphase and seems to be essential for maintaining a mobile lamin AC pool in the nuclear interior, which is required for proper nuclear functions [30]. A LAP2α missense polymorphism (Arg690Cys, rs17028450) was first identified in a patient with DCM and considered as a causal mutation, located in the C-terminal domain known to interact with lamin AC. In vitro studies have reported that this variant decreased the interaction of LAP2α with the lamin AC terminus [31]. This apparently functional variant was later found to be a frequent polymorphism in Latin American populations that does not cause DCM, although it was recently associated with an increased risk of neuromyelitis optica spectrum disorder in the Mexican population [32].

In the present study, we describe the family of a patient with early onset DCM, who was heterozygous for both a lamin AC truncating mutation (Ser431*) and the LAP2α Arg690Cys polymorphism. We modeled the wild-type 690Arg and the 690Cys LAP2α proteins, and their interactions with lamin AC to provide further evidence of its possible functional effects and used confocal microscopy to compare morphological parameters of leukocyte nuclei from family members with different *LMNA* and *TMPO* genotype combinations to explore whether this variant can be considered a genetic modifier of DCM in patients with *LMNA* mutations.

## 2. Results

The index case was a 32-year-old male with a positive family history of DCM (Figure 1). His father and two siblings were affected, and had passed away at ages 60, 24, and 25 years, respectively. He had 3 asymptomatic offspring aged 18, 13, and 8 years. The patient had a history of progressive dyspnea and orthopnea and was assessed as New York Heart Association (NYHA) Functional Class II on the first medical evaluation at INCICH. The electrocardiogram (ECG) showed a complete atrioventricular block and a left branch bundle block. Echocardiography reported reduced left ventricular ejection fraction (LVEF 22%), tricuspid annular phase systolic excursion (TAPSE) 33 mm, ventricular dilatation, and diffuse LV hypokinesis. MRI revealed dilated cardiomyopathy with a non-ischemic pattern, extended fibrosis, biventricular systolic dysfunction, tricuspid failure, and mild posterior pericardial effusion. Coronary arteries were normal on angiotomography. The patient presented with rapidly progressive congestive heart failure and arrhythmia requiring pacemaker implantation and died two years later at age 34. His 3 offspring were asymptomatic at the time, with normal ECGs and no signs of ventricular dilatation on echocardiography.

### 2.1. Molecular Diagnosis

Next-generation sequencing revealed that the index case was heterozygous for a truncating *LMNA* mutation (NM_001282625 c.1292C>A, p.Ser431*) and for the rs17028450 *TMPO* polymorphism (NM_003276:exon4:c.C2068T:p.Arg690Cys) (Appendix A). No other relevant variants were found in DCM-associated genes. The *LMNA* Ser431* mutation generates a premature stop codon at residue 431, truncating both lamin A and C proteins, and is pathogenic according to AMGC criteria (PVS1, PP5, PM2). Both variants were screened in available relatives by Sanger sequencing, revealing that all three offspring were *LMNA* p.Ser431* heterozygous. The heterozygous *TMPO* Arg690Cys polymorphism was also present in one of the offspring (IV-2), in a paternal uncle (II-1), and in 3 sisters of the index case (III-1, III-2, and III-6, Figure 1). All sisters without the lamin AC mutation who were heterozygous for the LAP2α polymorphism were asymptomatic and had normal echocardiograms.

A single Western blot experiment was performed on purified leukocyte extracts from each selected individual according to genotype combinations heterozygous for LAP2α Arg690Cys (case III-6), heterozygous for Lamin AC Ser431* (case IV-1), double heterozygous LAP2α Arg690Cys and Lamin AC Ser431* (case IV-2), and wild type (case III-5). As compared to leukocytes from the wild-type individual, Lamin A and C protein expression showed a sharp decrease in Lamin AC Ser431* heterozygous leukocytes (44% and 45%, respectively), and in Lamin AC Ser431*/LAP2 Arg690Cys compound heterozygous leukocytes (62% and 54%, respectively; Appendix A).

### 2.2. Protein Modeling Predicts the LAP2α Arg690Cys Polymorphism Destabilizes the LAP2α Homodimer

Homology modeling showed that one of the main features of the LAP2α homodimer is a long four-helix bundle formed by two helices from each monomer. Additional contacts between the shorter helical elements complete the interfacial region in which the R690 residue is located (Figure 2A,B). The presence of a cysteine residue in this position disrupts an ionic cluster of salt bridges normally formed between K689 and the R690 of one monomer, and three carboxylic sidechains from amino acids D537, D538, and E539 of the other monomer (Figure 2C). Figure 2D represents the four-helix bundle along the structure. This electrostatic contact seems to be structurally essential since it is not only conserved in the murine template but also virtually conserved in the whole family of homologs when a multiple sequence alignment is built. Thus, Arg690Cys is predicted to significantly change the structural stability of the dimer (Figure 2C).

Figure 3 shows all possible dimers that can be formed by Arg690Cys heterozygosity: dimers formed by two C690 chains, two wild-type R690 chains, and one 690Cys with one wild-type chain (Figure 2A–D). This analysis suggests that the LAP2α Arg690Cys variant destabilizes the LAP2α homodimer.

### 2.3. Effect of LAP2α Arg690Cys on LAP2α and Lamin AC Recognition in Silico

The human lamin AC model shows a long helical region followed by a LAP2α interaction domain of approximately 145 amino acids, and an immunoglobulin type domain (data not shown). The model suggests that the long helix can dimerize with another lamin AC chain, while the C-terminus domain recognizes LAP2α. Both structures are needed for the computational coupling of the lamin AC/LAP2α complex. The docking of the lamin AC binding domain on the surface of the LAP2α Arg690Cys heterodimer clearly showed an alteration of the two otherwise symmetrical interaction sites (Figure 4A,B). Although the 690 residue is not close to the lamin AC recognition site by affecting the LAP2α homodimer stability, the polymorphism is predicted to affect lamin AC and LAP2α binding.

A single LAP2α/Lamin AC co-immunoprecipitation experiment was performed in each patient selected according to genotype. When compared with wild-type leukocytes, co-immunoprecipitation of LAP2α with lamin A showed a 66.4% decrease in LAP2α Arg690Cys heterozygous leukocytes, a 97.7% decrease in lamin A/C Ser431* heterozygous leukocytes, and was imperceptible in double heterozygous leukocytes. Co-immunoprecipitation of LAP2α with lamin C showed a 45.3% decrease in LAP2α Arg690Cys heterozygous leukocytes, a 94.1% decrease in lamin A/C Ser431* heterozygous leukocytes, and was almost imperceptible in double heterozygous leukocytes (99% decrease, Appendix A).

### 2.4. Immunostaining Analysis

Leukocyte nuclei from patients bearing the Lamin AC Ser431* mutation (heterozygous and compound heterozygous) showed the characteristic blebs, invaginations, and lobulations on confocal microscopy in similar proportions for heterozygous (15–30% nuclei with irregularities) and compound heterozygous genotypes (16–28% nuclei with irregularities). Notably, blebs or invaginations were not observed in Lap2α Arg690Cys heterozygous nuclei. However, in this genotype, very few partial lobulations were apparent (Figure 5).

Nuclear shape descriptor measurements revealed that nuclei from Lamin AC Ser431* heterozygous leukocytes were significantly more elongated and had a smaller area compared to WT leukocyte nuclei, and although the mean circularity was lower than that of WT nuclei, the difference was not statistically significant. On the other hand, LAP2/Arg690Cys heterozygous nuclei showed significantly lower perimeter and higher circularity measurements than those of wild-type, Lamin AC Ser431* heterozygous, and Lamin AC Ser431*/LAP2α Arg690Cys double heterozygous nuclei, while area and elongation were similar in LAP2α Arg690Cys heterozygous and wild-type nuclei. The heterozygous presence of the LAP2α Arg690Cys polymorphism seemed to attenuate the effect of the Lamin AC truncating variant on the area, which was significantly higher in double heterozygous nuclei but remained significantly lower than in wild-type nuclei. Moreover, LAP2α heterozygosity seemed to compensate for the effect of the truncating variant on elongation, which was significantly lower in double heterozygous nuclei than in Lamin AC Ser431* nuclei and was similar to the elongation of wild-type nuclei (Figure 6).

LAP2α, Lamin AC, and DNA fluorescence signals were then measured through nuclei cross-sections, attempting to obtain a more objective measure of nuclear LAP2α and Lamin AC fluorescence distribution (Figure 7). LAP2α mean peak fluorescence was significantly higher in LAP2α Arg690Cys heterozygous, Lamin AC Ser431* heterozygous, and double heterozygous nuclei compared to wild-type nuclei (*p* < 0.01, *p* < 0.05 and *p* < 0.05, respectively). Moreover, lamin AC mean peak fluorescence was higher in Lamin AC Ser431* heterozygous and double heterozygous nuclei as compared to wild-type nuclei (*p* < 0.001 and *p* < 0.05, respectively). Mean ToPro fluorescence peaks did not differ among the different genotypes.

## 3. Discussion

Familial DCM is known for its genetic heterogeneity, incomplete penetrance, and variable expressivity, even among individuals bearing the same mutation. Although the factors modifying the expression and penetrance of familial DCM are not fully understood, different types of evidence suggest that some genetic variants may modify DCM expression severity [4,5,6]. The identification of a DCM family where a truncating *LMNA* mutation and a LAP2α missense polymorphism segregate independently represents an opportunity to study whether the LAP2α variant may modify disease expression.

The Lamin AC Ser431* mutation was previously identified in two Polish brothers with DCM [33], and is pathogenic according to ACMG criteria. Haploinsufficiency caused by premature-termination codon *LMNA* mutations is a known mechanism that can lead to cardiomyocyte disruption and underlie the pathogenesis of DCM [12,13,14,15,16,17,18]. *LMNA*^+/−^ heterozygous knockout mice showed 50% normal cardiac lamin AC levels, along with age-dependent development of atrioventricular nodal myocyte abnormalities (nuclear shape alterations and active apoptosis), AV conduction defects, impaired cardiomyocyte contractility, and DCM, revealing that lamin AC haploinsufficiency causes DCM with conduction system disease in mice [34,35]. Moreover, nonsense-mediated decay of the messenger (NMD) and haploinsufficiency have been proposed to explain the cardiac phenotype caused by large lamin AC deletions and other truncating mutations [18,36,37]. In this regard, Al Saaidi et al. used quantitative PCR, sequencing, and Western blot analyses to identify that the mutant *LMNA* Arg321* allele was degraded by nonsense-mediated mRNA decay in fibroblasts from heterozygous patients, and suggested that skewing of the lamin A to lamin C ratio may contribute to the phenotype [37]. Moreover, Cai et al. found nuclear blebs, accelerated nuclear senescence, and apoptosis of induced pluripotent stem cell-derived cardiomyocytes from *LMNA* Arg225Ter heterozygous patients [38]. The efficiency of NMD seems to be tissue-dependent [36] and there is evidence that other mechanisms are involved, such as apoptosis of cardiac conduction cells and mislocalization of mutant lamin to the endoplasmic reticulum and ER stress [34,35]. Although in the present study only one Western blot experiment was performed per patient, those bearing the Lamin AC Ser431* mutation (Lamin AC Ser431* heterozygous and Lamin AC Ser431*/LAP2α Arg690Cys double heterozygous) showed reduced Lamin AC expression in leukocytes (<50%), and no truncated protein was observed, which is consistent with haploinsufficiency in this cell type (Appendix A).

Leukocytes from both our lamin AC Ser431* patients showed the characteristic blebs, invaginations, and honeycomb images previously described in laminopathies, including DCM [12,13,14,15,16,17,18], even though both lamin AC Ser431* mutation carriers had not yet developed symptoms or echocardiographic evidence of DCM. As nuclear envelope lamin, AC proteins polymerize into complex filamentous meshworks associated with the internal nuclear membrane being responsible for nuclear membrane assembly, structure, and shape and providing protection from deformations caused by mechanical stress [19]. *LMNA* mutations or altered ratios of different lamin types caused by haploinsufficiency would disrupt this meshwork, affecting the structure of the nuclear envelope, and thus causing nuclear morphology abnormalities. Few studies have made quantitative measurements of nuclear morphology in DCM patients bearing *LMNA* mutations. Ferradini et al. (2021) [11] reported increased area and elongation but decreased circularity in leukocyte nuclei from DCM patients bearing *LMNA* missense mutations (Arg189Gln and Glu317Lys). Consistently, our DCM patient with the Ser431* mutation also showed nuclei with decreased circularity and increased elongation; however, in contrast with the report in missense mutations, nuclear area measurements were higher than in wild-type nuclei (*p* < 0.001). Further studies in patients with other truncating mutations may establish whether this difference is a characteristic of truncating mutations or haploinsufficiency.

Taylor and collaborators first identified LAP2α Arg690Cys in a European patient. Using in vitro binding assays, recombinant 690Cys LAP2α showed decreased interaction with the pre-lamin AC terminus in HeLa cells, and the authors suggested this variant was causal of DCM [31]. We found several lines of evidence that are consistent with the functional effect of this variant found in HeLa cells. First, our protein model predicted that the LAP2α 690Cys variant destabilizes the 690Cys homodimer. While the interchain salt bridge between residue 690 and carboxylates on residues 538 and 539 stabilizes the WT LAP2α homodimer formation, this interaction is lost in the 690Cys variant, apparently affecting the whole interface and lamina recognition. Additionally, the docking experiments predicted impaired interaction of the 690Cys LAP2α homodimer with lamin A/C. Although the co-immunoprecipitation results also suggest decreased interaction between LAP2α and lamins A and C, because only one co-immunoprecipitation experiment was performed per patient, further in vitro experiments are needed to confirm this finding. Second, the LAP2α heterozygous patient showed nuclear morphology changes in leukocytes (significantly decreased perimeter and increased circularity as compared to wild-type, Lamin AC Ser431* heterozygous, and double heterozygous nuclei), and the peak analysis revealed a more irregular distribution of LAP2α throughout the nucleus as compared to wild-type nuclei. Thus, using several different experimental approaches, we found additional evidence of the functional consequences of this variant. However, it is currently known that LAP2α Arg690Cys (rs17028450) is a frequent polymorphism in Latin American populations, with minor allele frequencies of 14% in Peruvians, 11% in Mexicans, and much lower frequencies in Asian, African, and European populations (<0.2%) (1000genomes.org, accessed on 26 October 2022) [39]. Thus, these functional effects are clearly not enough to cause disease.

Lamin A interacts with LAP2α in the nuclear interior, where they regulate euchromatic regions of the genome, affecting gene expression and possibly chromatin accessibility [30,40]. Recent experimental evidence suggests that the binding of LAP2α to lamin AC during interphase inhibits the formation of higher-order structures essential to maintaining a mobile lamin AC pool in the nuclear interior, which is required for proper nuclear functions [30]. While we have no direct evidence of the LAP2α 690Cys variant attenuating LAP2-mediated inhibition of higher order structures in the nucleoplasm, it is noteworthy that immunofluorescence peak analysis revealed that LAP2α distribution was more irregular in leukocytes from LAP2α Arg690Cys heterozygous patients as compared to the wild-type individual. More importantly, the presence of this variant seemed to attenuate the effect of Lamin AC Ser431* on the area of leukocyte nuclei, and to fully compensate for the effect of this mutation on nuclei circularity, suggesting it may be one of the many factors contributing to phenotype variation. However, the mechanisms of these effects remain to be determined.

This study has some limitations that should be pointed out. Blood samples were drawn from the participants only once. Thus, we could perform only a limited number of assays. Moreover, lamin AC immunostaining showed medium to low efficiency, and only a limited number of double-immunostained nuclei were available for analysis. However, the statistical power to identify significant differences in mean peak fluorescence was 0.88 and 0.94 for comparisons of LAP2α Arg690Cys heterozygous and lamin AC Ser431* heterozygous nuclei with wild-type nuclei. Additionally, we are unaware of the possible effect of cofounding factors, such as age, sex, or the presence of other genetic variants on leukocyte nuclear morphology. However, it is noteworthy that characteristic blebs, lobulations, and honeycomb images were present in the siblings bearing the lamin AC mutation before developing clinical symptoms. Finally, because of the young age of the patients, we are currently unaware of whether these morphological variations have a clinical translation. While we observed modifying effects of LAP2α Arg690Cys on nuclear morphology alterations caused by a lamin AC truncating mutation, the effects on the clinical phenotype were not evident. The double heterozygous index case had severe atrial and ventricular arrhythmia and heart failure at age 32 and died two years later. However, his father, inferred to be also double heterozygous, apparently had a milder phenotype and died at age 62. Unfortunately, he never attended the National Institute of Cardiology in Mexico City, and his clinical charts were unavailable. The 3 affected offspring of the patient are young and are still asymptomatic, but they represent an opportunity to carefully follow the clinical evolution of *LMNA*-associated DCM in the presence and absence of this LAP2α variant. Thus, although we have experimental evidence of a modifying effect of Arg690Cys on lamin AC-deficient nuclei morphology phenotypes, we currently do not know if this has a clinical translation, and the possible mechanisms responsible for these effects remain to be determined.

In conclusion, protein modeling, leukocyte nuclear morphology, and lamin AC/LAP2α fluorescence peak analysis suggest that the LAP2α Arg690Cys polymorphism is a functional polymorphism and may be one of the many factors modifying the clinical phenotype of patients with *LMNA*-associated dilated cardiomyopathy. Further studies are required to assess the possible clinical translation of these findings.

## 4. Materials and Methods

### 4.1. Subjects and Clinical Evaluation

A Mexican patient with a positive family history of dilated cardiomyopathy and sudden death attending the Heart Failure Clinic at the Instituto Nacional de Cardiología “Ignacio Chávez” (INCICH) was referred to the Instituto Nacional de Medicina Genómica in Mexico City for molecular diagnosis. All available first- and second-degree relatives were invited to participate in the study and submitted to a detailed medical evaluation, including echocardiographic analyses, according to current guidelines [41]. Informed consent and assent were obtained from all participating family members or their legal guardians, and the study was approved by the Review Board and the Ethics Committee of the Instituto Nacional de Medicina Genómica (Code 08/2016/I).

### 4.2. DNA Analysis

Genomic DNA was extracted from peripheral blood samples using the DNeasy Midi Blood kit (Qiagen, Germantown, MD, USA). DNA from the index case was sequenced using site-directed next-generation sequencing (TruSight Cardio, Illumina, San Diego, CA, USA) on a NextSeq device. Post-run sequencing quality was assessed using FastQC software version 0.11.9 (Babraham, Bioinformatics, UK), alignment and variant calling were performed with BWA Enrichment v2.1.0 on the BaseSpace platform (Illumina), and variants were annotated with Annovar software (wnnovar.wglab.org, Wang Genomics Lab, Philadelphia, PA, USA, accessed on August 12, 2020 [42]. Variants of interest were interpreted classified according to the guidelines of the American College of Medical Genetics and Genomics refined for DCM [43], and screened in all participating family members by Sanger sequencing.

### 4.3. Protein Modeling

The human alpha-thymopoietin isoform sequence (694 amino acids, accession number NP_003267.1) was used for modeling purposes. BLASTp, as implemented in blast.ncbi.nlm.nih.gov (accessed on 26 October 2022), was used for sequence-homology searches, while Molecular Oriented Environment (MOE, www.chemcomp.com, accessed on 10 August 2021) was used for molecular modeling, editing, visualization, and structural analysis. For wild-type (WT) alpha-thymopoietin modeling, a homodimeric structure of the murine LAP2α C-terminal domain was selected, namely the 2V0X file from Protein Data Bank, rcsb.org (PDB). To complete missing atoms and segments of the main chain, minor adjustments were performed by superposing a copy of chain B on chain A. Initially, one hundred structures were generated, which differed in the inserted side chain conformations and added loops. Afterwards, each structure was locally minimized until an average force of 0.5 kcal/(mol Å) was obtained. The best-packed structure obtained according to MOE folding parameters was then globally minimized to 0.01 kcal/(mol Å) to yield the final WT model. For single and double mutant dimers, the resulting human WT structure was used as a starting model, where arginine 690 was replaced by cysteine, including the 25 initial models. Crystallographic water molecules were included during modeling in all cases. All energy calculations were performed using the CHARMM27 force field [44].

The structure of human lamin AC expressed in E. coli, determined by NMR [45] was downloaded from PDB file 1IVT. This structure corresponds to the C-terminus domain containing an Ig folding pattern. The docking of lamin AC into the structure of the modeled LAP2α dimer was performed using Hex, ZDOCK, and HADDOCK servers [46,47,48].

### 4.4. Leukocyte Purification and Immunofluorescence (IF) Analysis

Leukocytes were purified from peripheral blood samples of selected participants (cases III-5, III-6, IV-1 and IV-2) according to Vacutainer^®^ CPT (BD Bioscience, Franklin Lakes, NJ, USA) procedure instructions [49,50]. Purified leukocytes were fixed with 4% formaldehyde in PBS (pH 7.4) for 10 min at room temperature, washed three times, and incubated for 1 h with a washing solution [0.25% Triton X-100 in PBS (PBST)]. Leukocytes were blocked with 1% BSA in PBST for 30 min at room temperature. Rabbit lamin AC antibody (1:100), or Mouse LAP2α antibody (1:50) were incubated overnight at 4 °C in blocking solution. Cells were washed three times and incubated for 1 h at room temperature in the dark with either Alexa Fluor 568 donkey anti-rabbit antibody (1:2000) or Alexa Fluor 488 donkey anti-mouse antibody (1:2000) (Abcam, Cambridge, UK) for lamin AC and LAP2α detection. Nuclear DNA was stained with ToPro (1:500) (Life Technologies, Carlsbad, CA, USA) for 30 min at room temperature. Finally, the cells were rinsed, drained, and fixed using Prolong Gold antifade reagent (Life Technologies). Pictures were captured using confocal microscopy (Leica SP8-DM6000/LasAF, Leica Microsystems, Wetzlar, Germany) and images were analyzed with Fiji software (ImageJ v1.53t, National Institutes of Health, Bethesda, MD, USA).

Nuclei shape descriptors (area, perimeter, circularity, and elongation) were estimated in leukocytes purified from participants with different genotypes: heterozygous for LAP2α Arg690Cys (case III-6, 266 nuclei), heterozygous for Lamin AC Ser431* (case IV-1, 126 nuclei), double heterozygous LAP2α Arg690Cys and Lamin AC Ser431* (case IV-2, 523 nuclei), and wild type (case III-5, 184 nuclei). To obtain a more objective measure of protein distribution throughout the nucleus, lamin AC, LAP2α and ToPro fluorescence was measured along cross-sections of leukocyte nuclei from individuals with all genotype combinations, and mean peak fluorescence from all signals were compared among different genotypes with the Peak Analyzer Tool (OriginPro 2020, Origin Lab, North Hampton, MA, USA). Data are expressed as mean ± SEM. The significance of the differences between means was assessed with one-way ANOVA, followed by post-hoc Tukey’s tests of significance.

### 4.5. Protein Extraction and Immunoblot Analysis

Total protein extracts were prepared from purified leukocytes [51] using a lysis buffer containing 50 mM Tris (pH 8), 150 mM NaCl, 200 mM EDTA, 100 mM NaF, 10 mM sodium pyrophosphate, 2 mM Na3VO4, 1 mM phenylmethylsulfonyl fluoride, and 0.5% (*v*/*v*) Nonidet 40. Lysates were incubated on ice for 30 min and centrifuged at 14,000× *g* for 10 min at 4 °C, as previously reported [52]. The buffer was supplied with 1:10 diluted protease and phosphatase inhibitor cocktail according to the manufacturer’s instructions (Roche, Branchburg, NJ, USA). All procedures were performed at 4 °C to reduce protein degradation. Equivalent amounts of boiled protein in Laemmli’s buffer were analyzed through SDS-PAGE and transferred to a PVDF membrane. Anti-Lamin AC antibody (NBP2-25151SS, Novus Biologicals, Centennial, CO, USA) and anti-GAPDH antibody (ab181603, Abcam, Cambridge, UK) were used for immunoblotting. Protein loading was confirmed by reprobing the blots with anti-GAPDH. Densitometric analyses were carried out using Fiji software (NIH).

### 4.6. Immunoprecipitation (IP) Analysis

Lysates from purified leukocytes of peripheral blood were solubilized in ice-cold lysis buffer containing 20 mM Tris-Cl, pH 7.8, 137 mM NaCl, 1% (*v*/*v*) Nonidet P40, 10% (*v*/*v*) Glycerol, and 2 mM EDTA supplied with 1:10 diluted protease and phosphatase inhibitor cocktails (Roche, Branchburg, NJ, USA). Five hundred micrograms of total protein were incubated at 4 °C overnight with 2 μg of human anti-LAP2 antibody. Protein A/G-Agarose complex beads (20422, Thermo Scientific™. Waltham, MA, USA) were then added and incubated for 4 h at 4 °C. The beads were washed four times with lysis buffer. Immune complexes were eluted by boiling in Laemmli’s buffer and centrifuged; proteins were analyzed by immunoblot analysis for Lamin AC detection.

## Figures and Tables

**Figure 1 ijms-23-13626-f001:**
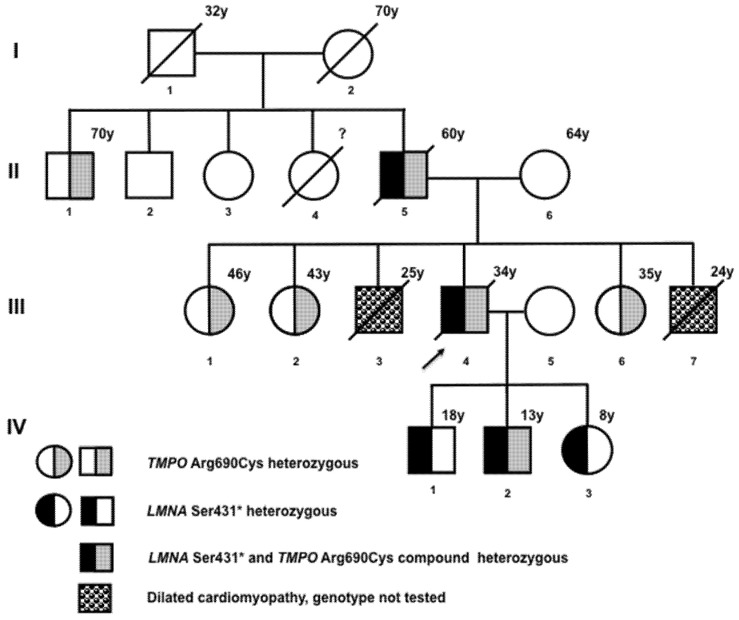
Family Pedigree. The arrow points to the proband on whom next-generation sequencing was performed. Squares and circles half-filled in black are heterozygous *LMNA* Ser431* mutation carriers, squares, and circles half-filled in gray are heterozygous *TMPO* Arg690Cys polymorphism carriers. Individuals II-5, III-3, III-4, and III-7 had clinical manifestations of dilated cardiomyopathy; the genotype was not tested in individuals III-3 and III-7.

**Figure 2 ijms-23-13626-f002:**
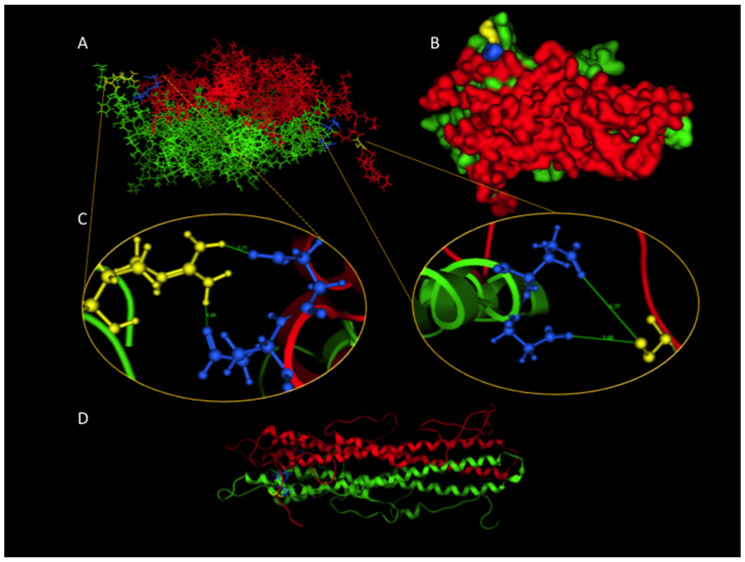
Molecular modeling of the LAP2α dimer. (**A**) Graphical representation of LAP2α heterodimer formed by one wild-type (green) and one 690Cys (red) subunit. Side chains in position 690 are depicted in yellow, and nearby aspartate 538 and glutamate 539 residues from the other subunits are depicted in blue. These residues form an interchain electrostatic contact in the wild-type form. (**B**) Heterodimer view rotated about 90° around the horizontal and perpendicular axes. (**C**) Zoom to both interchain contacts involving residue 690. The left panel shows the wild-type arginine forming ionic pairs with aspartate and glutamate, and electrostatic interchain contacts. The right panel shows the modeled interface with the Arg690Cys polymorphic chain, resulting in the loss of interchain contact, allowing the other chain to move apart during modeling. (**D**) Lateral view of the LAP2α dimer in ribbon representation to emphasize interface contact and dimer symmetry.

**Figure 3 ijms-23-13626-f003:**
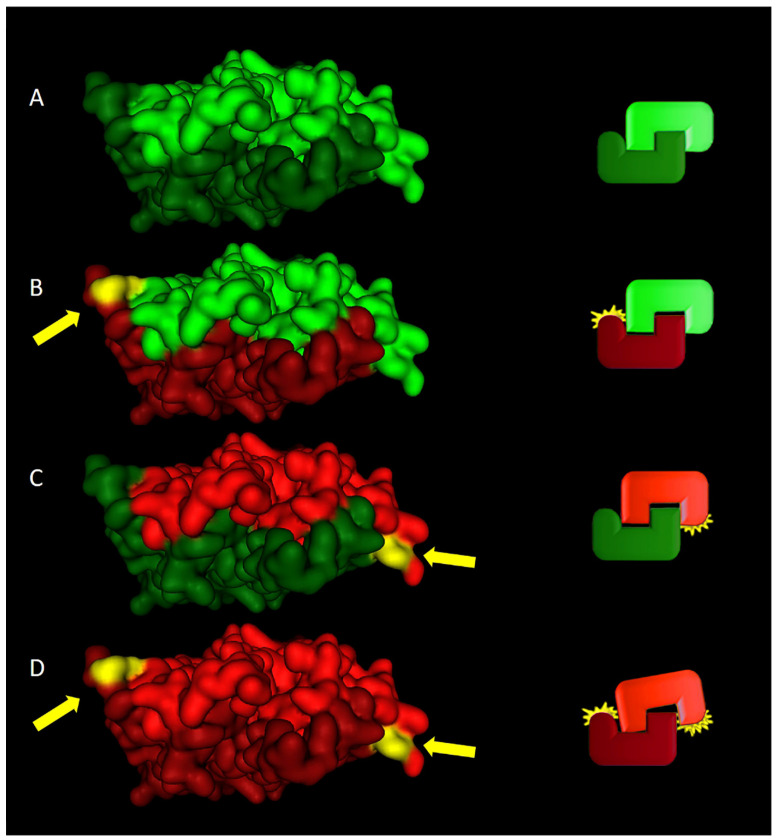
Different assembly scenarios of the LAP2α R690C dimer in heterozygosity. (**A**) LAP2α wild-type homodimer assembly. (**B**,**C**) LAP2α Arg690Cys heterodimer assembly. (**D**) LAP2α 690Cys homodimer assembly. The left panel shows the dimer model, and the right panel represents a graphical view of different assembly scenarios showing 690Cys homodimer instability. The wild-type 690Arg chain is depicted in green; the 690Cys chain is depicted in red. Arrows indicate the 690Cys residue.

**Figure 4 ijms-23-13626-f004:**
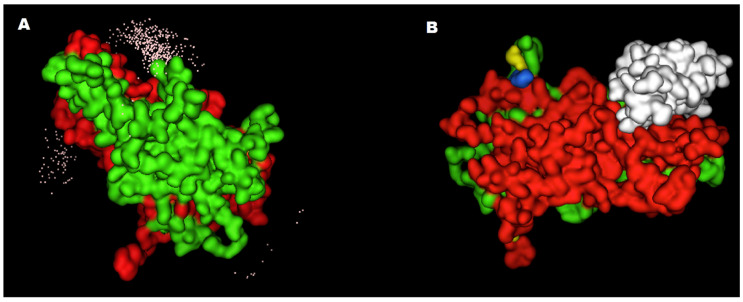
Lamin AC-LAP2α docking. (**A**) White dots show the best five hundred predicted docking positions of the lamin AC binding domain on the LAP2α Arg690Cys heterodimer surface, according to in silico assays. Each dot represents the mass center of one of the predicted positions of the lamin AC docked on LAP2α. Dots set on the left side must be equivalent positions to those depicted on the top. However, this heterodimer shows a differential recognition preference because of the effects of the variant. (**B**) The best docking of lamin AC on the surface of the LAP2α dimer. The wild-type LAP2α chain is depicted in green and the 690Cys chain is depicted in red. Cysteine 690 is represented in yellow, and aspartate 538 and glutamate 539 are represented in blue. The LAP2α recognition domain of lamin A/C is depicted in white.

**Figure 5 ijms-23-13626-f005:**
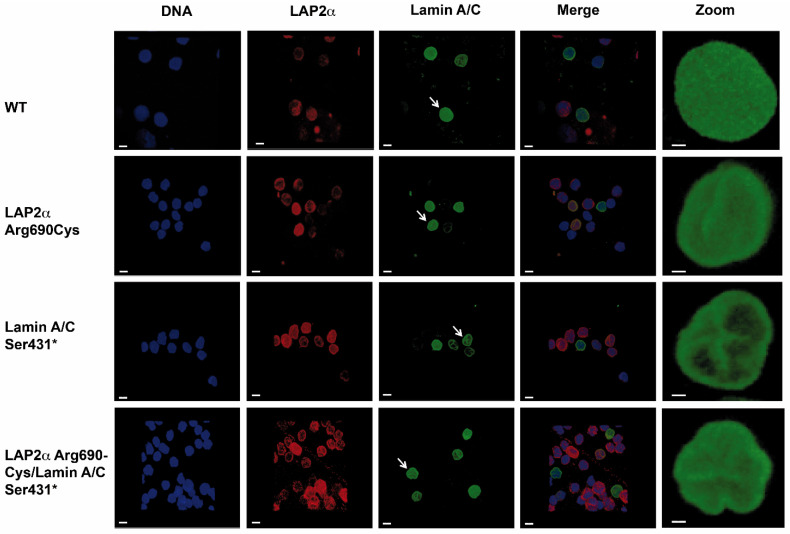
Confocal microscopy showing leukocyte nuclei from family members with different genotypes. Immunofluorescence staining shows blebs, invaginations, and honeycomb images only in nuclei from patients bearing the Lamin AC Ser432* mutation (heterozygous and compound heterzogyous). White arrows indicate the cells magnified in the right column.

**Figure 6 ijms-23-13626-f006:**
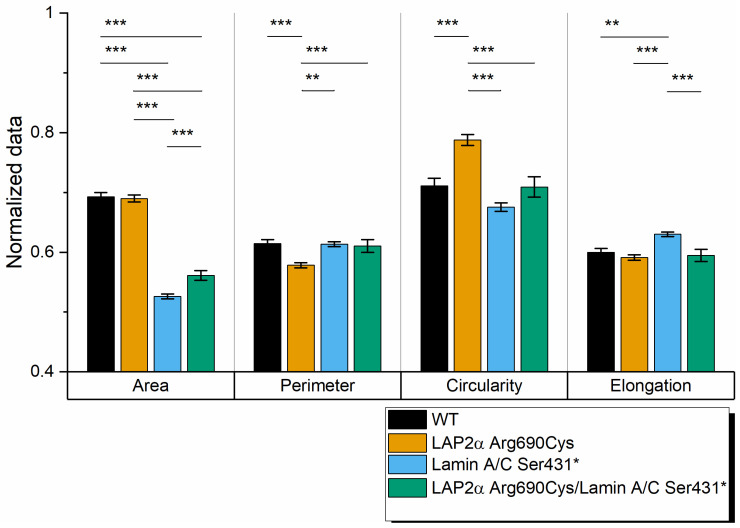
Comparison of shape descriptors for leukocyte nuclei from family members with different Lap2α Arg690Cys and Lamin AC Ser431* variant combinations. Bar graphs report the mean ± SEM values of each descriptor. Differences between genotypes were tested with one-way ANOVA and post hoc Tukey’s mean comparisons. ** p <* 0.05; ** *p* < 0.01, *** *p* < 0.001.

**Figure 7 ijms-23-13626-f007:**
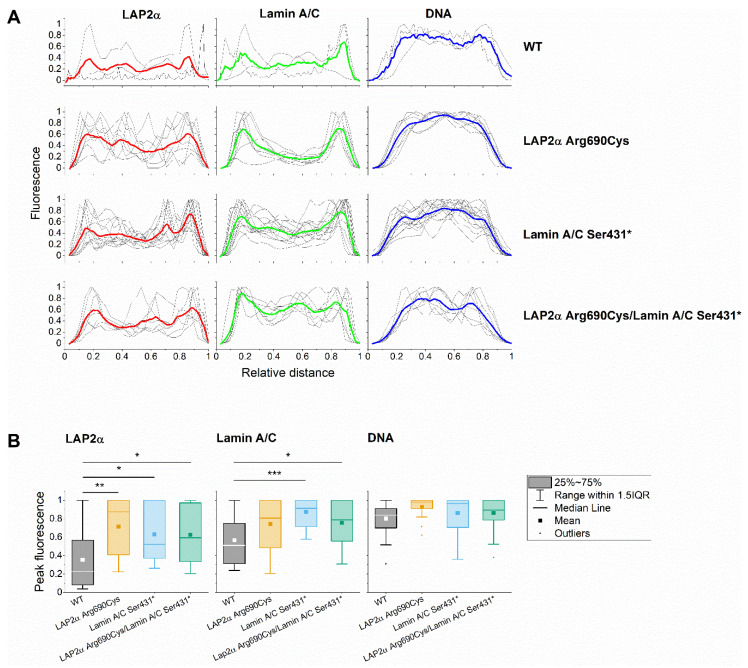
Mean peak fluorescence across leukocyte nuclei according to genotype. (**A**) Relative Lap2α, Lamin AC, and DNA fluorescence signal intensity profiles along leukocyte nuclei cross-sections from individuals with different genotypes. Each gray line represents the profile of one nucleus, and the colored lines represent mean fluorescence intensities. (**B**) Box plots comparing Lap2α, Lamin AC, and DNA mean peak fluorescent values in different genotypes. * *p* < 0.05; ** *p* < 0.01, *** *p* < 0.001. The analysis was performed using the images from Figure 5.

## Data Availability

Not applicable.

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
