# Peer review of "Leukocyte Nuclear Morphology Alterations in Dilated Cardiomyopathy Caused by a Lamin AC Truncating Mutation (LMNA/Ser431*) Are Modified by the Presence of a LAP2 Missense Polymorphism (TMPO/Arg690Cys)"

_ijms, 2022, doi:10.3390/ijms232113626_

Round 1

Reviewer 1 Report

The authors try to find the role of LMNA/Ser431* and TMPO/Arg690Cys (2 possible pathogenic variant) on leukocyte nuclear morphology alteration in dilated cardiomyopathy. I have many concerns.

1. It is not clear why the authors focus on the leukocyte nuclear morphology alteration in dilated cardiomyopathy. How does leukocyte nuclear morphology alteration cause dilated cardiomyopathy? This is required to be well clarified.

2. It is still not known whether LMNA Ser431* can cause DCM, as all the 3 offspring (IV-1, -2 and -3) don’t have dilated cardiomyopathy. Direct evidence required to be provided to show LMNA Ser431* is able to cause DCM.

3. Even whether TMPO/Arg690Cys is able to cause DCM is still not clear. More evidence is required to be provided. Do II-1, III-1, III-2 and III-6 have DCM?

4. The author found that LAP2α Arg690Cys apparently attenuated the effect of LMNA Ser431* on nuclear area, and fully compensated its effect on nuclear circularity. Does this mean LAP2α Arg690Cys is a therapeutic mutation from the authors’ results? In addition, does this mean the leukocyte nuclear morphology alteration contributes nothing to DCM? These are required to be well clarified.

5. The author concluded in the end of the Abstract that the data suggest LAP2α/Arg690Cys may be one of the many factors contributing to phenotype variation of LMNA-associated DCM. How could the authors reach such conclusion from the results that LAP2α Arg690Cys apparently attenuated the effect of LMNA Ser431* on nuclear area and fully compensated its effect on nuclear circularity?

6. In Figure5, why is the expression level of LAP2a in WT lower than the other 3 groups?

7. Also in Figure5, the percentages of the nuclei with characteristic blebs, invaginations and lobulations in all 4 groups are required to be quantified.

Author Response

RESPONSE TO REVIEWER 1

Thank you for taking the time to review our manuscript and for your observations and suggestions.

The authors try to find the role of LMNA/Ser431* and TMPO/Arg690Cys (2 possible pathogenic variant) on leukocyte morphology alteration in dilated cardiomyopathy. I have many concerns.

Response:  We must clarify that finding the role of LMNA/Ser431* as a pathogenic variant in DCM is not one of the purposes of the study.  This variant is known to cause DCM because truncating LMNA mutations are a well-established cause of DCM.  We have clarified that this variant is pathogenic according to ACMG criteria and provide the criteria that allow to state that this mutation is  pathogenic (PVS1, PP5 and PM2).

Before responding to all concerns raised by the reviewer, we would like to state that we changed figure 3 to be consistent with the color code in other figures.

  1. It is not clear why the authors focus on the leukocyte nuclear morphology alteration in dilated cardiomyopathy. How does leukocyte nuclear morphology alteration cause dilated cardiomyopathy? This is required to be well clarified.

The mechanisms by which LMNA mutations lead to DCM are not fully understood and involve many processes such as nuclear membrane stability, gene expression, mechano-transduction, etc.  The introduction section states that the lamin meshwork provides protection from deformations caused by mechanical stress (page 2, line 72). Altered leukocyte morphology is characteristic of all laminopathies, including those that do not cause DMC. In this family, these characteristic morphological alterations were evident before the clinical onset.  Thus, this allowed us to assess possible pre-clinical phenotypic manifestations in our patients.  We have clarified this in the manuscript.

  1. It is still not known whether LMNA Ser431* can cause DCM, as all the 3 offspring (IV-1, -2 and -3) don’t have dilated cardiomyopathy. Direct evidence required to be provided to show LMNA Ser431* is able to cause DCM.

Although DCM has a very high penetrance, this penetrance is age-related.  Most patients bearing causal mutations show signs or symptoms of the disease starting in adulthood, and the 3 offspring are expected to develop clinical manifestations later on. This has been clarified in the introduction section.  Moreover, LMNA Ser431* is a truncating mutation, and it is well established that heterozygous truncating LMNA mutations are causal of DCM.

  1. Even whether TMPO/Arg690Cys is able to cause DCM is still not clear. More evidence is required to be provided. Do II-1, III-1, III-2 and III-6 have DCM?

Although Taylor et al. described it initially as a mutation causal of DCM, it is currently known that the TMPO Arg690Cys variant does not cause DCM, it is a common polymorphism with a minor allele frequency >10% in Latin American populations.  Cases III-1, III-2 and III-6 (ages 46, 43 and 35 years) were assessed by cardiologists and do not have any clinical or echocardiographic signs of DCM.  We were not clear when we called this polymorphism a “variant”.  We have changed the term “variant” for “polymorphism” in several instances to avoid this confusion, and have clarified that cases III-1, III-2 and III-6 showed no clinical or echocardiographic signs or symptoms of DCM in the results section.

  1. The author found that LAP2α Arg690Cys apparently attenuated the effect of LMNA Ser431* on nuclear area, and fully compensated its effect on nuclear circularity. Does this mean LAP2α Arg690Cys is a therapeutic mutation from the authors’ results? In addition, does this mean the leukocyte nuclear morphology alteration contributes nothing to DCM? These are required to be well clarified.

Because LAP2α Arg690Cys is a polymorphism, and because this protein is known to interact with Lamin A/C, the purpose of the study was to assess whether this polymorphism is a genetic modifier of a Mendelian form of DCM caused by Lamin A/C mutations. Because the offspring are still young, we are unable to assess whether there are clinical differences in patients with different genotype combinations.  However, it is noteworthy that the nuclear morphology alterations were present before the clinical manifestations, and there were statistically significant differences in leukocyte morphology parameters.  While this suggests that there may be clinical differences, we will have to wait several years to see if the differences do in fact have a clinical translation.  We have clarified this in the discussion section.

  1. The author concluded in the end of the Abstract that the data suggest LAP2α/Arg690Cys may be one of the many factors contributing to phenotype variation of LMNA-associated DCM. How could the authors reach such conclusion from the results that LAP2α Arg690Cys apparently attenuated the effect of LMNA Ser431* on nuclear area and fully compensated its effect on nuclear circularity?

Genetic modifiers of disease may increase the severity or attenuate the severity of Mendelian disease.  We state that it seems to attenuate the changes in nuclear morphology, but did not alter mean peak distribution, although we do not yet know if this will have a clinical translation because the patients have not yet developed DCM clinical manifestations.

  1. In Figure 5, why is the expression level of LAP2a in WT lower than the other 3 groups?

We repeated the immunostaining experiments several times, and the Lamin AC antibody had medium to low immunostaining efficiency, while the LAP2 antibody had medium to high efficiency. Because the family did not answer our attempts to reach them after the first blood samples were drawn, we were not able to repeat these experiments.   In spite of this technical problem, we were able to observe statistically significant differences among genotype combinations.  This low efficiency immunostaining problem has been acknowledged and included in a paragraph discussing the limitations of the study.

  1. Also in Figure 5, the percentages of the nuclei with characteristic blebs, invaginations and lobulations in all 4 groups are required to be quantified.

The percentage of cells with blebs, invaginations or lobulations was similar in heterozygous and compound heterozygous DCM individuals.  We have included the data in the results section. Because identifying partial lobulations or small irregularities can vary according to the perception of different observers, we used mean peak fluorescence analysis attempting to provide a more objective measure of these nuclear morphology irregularities.  Lobulations appear as peaks in the analysis graphs, and their presence increases the mean peak fluorescence. While there was no evidence supporting that LAP2 Arg690Cys modifies mean peak fluorescence, the presence of this variant may modify some, but not all nuclear morphology characteristics caused by LMNA mutations.

Reviewer 2 Report

1. Figure legend for Figure 1 is missing.

2. Index case is described as 32 years old in text (line 97) while Figure 1 shows 34 years old.

3. In Figure 5, why do some nuclei stain for LAP2a and some for only for Lamin? Why aren’t there more nuclei with double staining?

4. In Figure 6, what does normalized data in y-axis mean? Please explain why there could be changes in perimeter of LAP2a Arg690Cys but not the area of the nuclei in this genotype?

5. It is not clear how the mean peak fluorescence analysis is useful. The so called irregularities in all mutant genotypes differ significantly from wildtype and it does not really support that LAP2a Arg690Cys is a modifier of this phenotype.

Author Response

RESPONSE TO REVIEWER 2

Thank you for taking the time to review our manuscript, for your suggestions and observations.

Before responding to all concerns raised by the reviewer, we would like to state that we changed figure 3 to be consistent with the color code in other figures.

  1. Figure legend for Figure 1 is missing.

Thank you for the observation, we have included the legend.

  1. Index case is described as 32 years old in text (line 97) while Figure 1 shows 34 years old.

The index case was 32 when first assessed at the Instituto Nacional de Cardiología Ignacio Chávez.  We passed away at age 34, two years later as stated in the text. This is stated in the first paragraph of the results section.  

  1. In Figure 5, why do some nuclei stain for LAP2a and some for only for Lamin? Why aren’t there more nuclei with double staining?

We repeated the immunostaining experiments several times, and the Lamin AC antibody had medium to low immunostaining efficiency, while the LAP2 antibody had medium to high efficiency. Because the family did not answer our attempts to reach them after the first blood samples were drawn, we were not able to repeat these experiments.   In spite of this technical problem, we were able to observe statistically significant differences among genotype combinations.  This low efficiency immunostaining problem has been acknowledged and included in a paragraph discussing the limitations of the study.

  1. In Figure 6, what does normalized data in y-axis mean? Please explain why there could be changes in perimeter of LAP2a Arg690Cys but not the area of the nuclei in this genotype?

 In figure 6 data were normalized dividing each data point per the WT maximum value. Bars show the mean ± SEM. The circle is the shape with the largest area in the smallest perimeter, here, we are showing that LAP2a Arg690Cys produces more circular nuclei, a smaller perimeter with a conserved area is compatible with the former notion.  We have clarified this in the discussion section.

  1. 5. It is not clear how the mean peak fluorescence analysis is useful. The so called irregularities in all mutant genotypes differ significantly from wildtype and it does not really support that LAP2a Arg690Cys is a modifier of this phenotype.

We used mean peak fluorescence analysis attempting to provide a more objective measure of these nuclear morphology irregularities.  We found that the identification of small blebs, invaginations or small or partial lobulations is subjective and the identification of cells with milder morphological alterations varied among independent observers. Lobulations appear as peaks in the analysis graphs, and their presence increases the mean peak fluorescence. We agree that there is no evidence supporting that LAP2 Arg690Cys modifies mean peak fluorescence.  However, the presence of this variant may modify some, but not all nuclear morphology characteristics caused by LMNA mutations. We have clarified this in the text.

Round 2

Reviewer 1 Report

The authors don’t well address my concerns. Multiple issues remain outstanding.

1. The authors stated that altered leukocyte morphology is characteristic of all laminopathies, including those that do not cause DMC, which should be added to the manuscript together with the relevant references. In addition, the author should provide evidence that laminin-accociated DCM could cause leukocyte nuclear morphology alterations.

2. Potential mechanism should be provided about how LMNA Ser431* causes leukocyte nuclear morphology alterations.

3. It is still not clear whether LMNA Ser431* can cause DCM. Additional data are required to provide as direct evidence to show LMNA Ser431* is able to cause DCM.

Author Response

RESPONSE TO THE OBSERVATIONS OF REVIEWER 1, ROUND 2

 Thank you for taking the time to review our manuscript.

  1. The authors stated that altered leukocyte morphology is characteristic of all laminopathies, including those that do not cause DMC, which should be added to the manuscript together with the relevant references.

We have extended the information on different laminopathies in the introduction section, the OMIM reference for LMNA laminopathies has been added.

  1. In addition, the author should provide evidence that laminin-accociated DCM could cause lukocyte nuclear morphology alterations:

Previous studies have provided this evidence, and some of these studies are referred in the manuscript.  Nuclear morphology alterations are considered a hallmark of laminopathies.  We have extended this description and hope it is now clear. 

  1. Potential mechanism should be provided about how LMNA Ser431* causes leukocyte nuclear morphology alterations.

As stated above, the mechanisms are unclear and seem to involve many mechanisms that vary in different tissues and cells. We have extended this information describing how lamin AC haploinsuficiency may cause nuclear morphology alterations in the discussion section, and that the lamin meshwork provides structure to the nucleus, which is altered in case of lamin AC haploinsufficiency.

  1. It is still not clear whether LMNA Ser431* can cause DCM. Additional data are required to provide as direct evidence to show LMNA Ser431* is able to cause DCM.

There is no doubt that truncating LMNA mutations cause DCM, as several studies using different experimental models support this notion. lmna-/* knockout mice develop DCM providing direct evidence that heterozygous lamin haploinsufficiency can cause DCM in the mouse model.

We provide evidence that the Ser431* mutation causes haploinsufficiency in our patients.  We stated in the discussion section that the Western Blot analysis (supplementary figure 2) is compatible with haploinsufficiency  We have stressed this in the discussion section, and added that the Western Blot analysis shows no evidence of a truncated protein, as described by Al-Saaidi et al.

Round 3

Reviewer 1 Report

The authors don’t well address my concerns. Multiple issues remain outstanding.

1. Although the authors stated that nuclear morphology alterations are considered a hallmark of laminopathies, it is still not clear why the authors only focused on the leukocyte. The authors must provide direct evidence that altered leukocyte nuclear morphology is characteristic of all laminopathies and laminin-accociated DCM could cause leukocyte nuclear morphology alterations.

2. The author stated that LMNA/Ser431* could cause leukocyte nuclear morphology alterations in dilated cardiomyopathy. Potential mechanism must be provided about how LMNA Ser431* causes leukocyte nuclear morphology alterations instead of just saying the mechanisms are unclear.

3. Additional data must be provided as direct evidence to show it is LMNA Ser431* that is really able to cause DCM, as the author can’t provide evidence that IV-1, IV-2 and IV-3 have DCM.

Author Response

RESPONSE TO THE CONCERNS OF REVIEWER 2.

All concerns raised by reviewer 2 were considered in this revised version of the manuscript.

  1. "Although the authors stated that nuclear morphology alterations are considered a hallmark of laminopathies, it is still not clear why the authors only focused on the leukocyte. The authors must provide direct evidence that altered leukocyte nuclear morphology is characteristic of all laminopathies and laminin-accociated DCM could cause leukocyte nuclear morphology alterations."

Nuclear morphology alterations are indeed a hallmark of laminopathies, as previously established in the medical literature by many authors over the last decades. Moreover, as shown by Ferradini et al. and other authors (Liu et al., 2016; Wada et al., 2021), leukocytes are one of several cell types showing nuclear morphology alterations in DCM patients with LMNA mutations. We focus on the leukocytes because they can be observed using a blood sample. There is no need to submit the patients to more invasive methods such as skin biopsies or heart biopsies. Drawing a blood sample is non-invasive.

We have clarified that leukocytes are one of the cell types affected by nuclear morphology alterations in LMNA-associated DCM in the introduction section. The statement reads:

"Leukocytes are one of several cell types showing nuclear morphology alterations in DCM patients with LMNA mutations [10, 11]." 

  1. "The author stated that LMNA/Ser431* could cause leukocyte nuclear morphology alterations in dilated cardiomyopathy. Potential mechanism must be provided about how LMNA Ser431* causes leukocyte nuclear morphology alterations instead of just saying the mechanisms are unclear."

 We proposed a mechanism in the second revised version, discussion section, page 12 , lines 348-354. It states as follows:

“As nuclear envelope lamin AC proteins polymerize into complex filamentous meshworks associated with the internal nuclear membrane being responsible for nuclear membrane assembly, structure, shape, and providing protection from deformations caused by mechanical stress [17]. LMNA mutations or altered ratios of different lamin types caused by haploinsufficiency would disrupt this meshwork affecting the structure of the nuclear envelope, and thus cause nuclear morphology abnormalities.”

  1. "Additional data must be provided as direct evidence to show it is LMNA Ser431* that is really able to cause DCM, as the author can’t provide evidence that IV-1, IV-2 and IV-3 have DCM."

We do in fact provide evidence that IV-1 and IV-2 will most likely develop DCM:

  1. They have nuclei morphology alterations, which are a hallmark (as clearly established in the medical literature) of laminopathies. Nuclear morphology alterations do not occur in healthy individuals.
  2. They have LMNA haploinsufficiency in leukocytes, as shown in the Western Blot analysis (Supplementary Figure 2).
  3. LMNA Haploinsufficiency is a well-established mechanism causing DCM in an age-dependent manner, with evidence in animal knockout models and in individuals or families bearing truncating or premature stop codon mutations. This is all stated and referenced in the manuscript, both in animal models (knockout mice), and in families bearing other truncating LMNA (premature stop codon) mutations. This is all described in the Discussion section, pages 11-12, lines 318-338, and all appropriate references were provided. It reads as follows:

“Haploinsufficiency caused by premature-termination codon LMNA mutations is a known mechanism 320 that can lead to cardiomyocyte disruption and underlies the pathogenesis of DCM [10-16]. Lmna+/- heterozygous knockout mice showed 50% of normal cardiac lamin AC levels, along with age-dependent development of atrioventricular nodal myocyte abnormalities (nuclear shape alterations and active apoptosis), AV conduction defects, impaired cardiomyocyte contractility and DCM, revealing that lamin AC haploinsufficiency causes DCM with conduction system disease in mice [32, 33]. Moreover, nonsense-mediated decay of the messenger (NMD) and haploinsufficiency have been proposed to explain the cardiac phenotype caused by large lamin AC deletions and other truncating mutations [16, 34, 35]. In this regard, Al Saaidi et al. used quantitative PCR, sequencing and Western Blot analyses to identify that the mutant LMNA Arg321* allele was degraded by nonsense-mediated mRNA decay in fibroblasts from heterozygous patients, and suggested skewing of the lamin A to lamin C ratio may contribute to the phenotype [35]. Moreover, Cai et al. found nuclear blebs, accelerated nuclear senescence and apoptosis of induced pluripotent stem cell-derived cardiomyocytes from LMNA Arg225Ter heterozygous patients [36]. The efficiency of NMD seems to be tissue-dependent [34] and there is evidence that other mechanisms are involved, such as apoptosis of cardiac conduction cells and mislocalization of mutant lamin to the endoplasmic reticulum and ER stress [32, 33].”

4. Previous researchers provided ample evidence that LMNA haploinsufficieny causes DCM. We provide evidence that two of the patients bearing the mutation have LMNA haploinsufficiency. Page 12, lines 338-343. The statement reads as follows:

“Although in the present study only one Western blot experiment was performed per patient, those bearing the Lamin AC Ser431* mutation (Lamin AC Ser431* heterozygous and Lamin AC Ser431*/LAP2α Arg690Cys double heterozygous) showed reduced Lamin AC expression in leukocytes (<50%), and no truncated protein was observed, which is consistent with haploinsufficiency in this cell type (Supplementary figure 2)."

5. We understand that they have not yet developed DCM, but the expression of this disease is age-dependent, with full penetrance by age 60 years, and the patients are still young. Epidemiological studies that are cited in our manuscript state that the penetrance is complete at age 60. (Page 2, lines 63-65). The eldest offspring with the mutation is only 18 years old, he has no symptoms yet, but already has nuclei morphology alterations in leukocytes. 

The age-related penetrance statement reads as follows: 

“The development of the phenotype occurs between ages 20-39 years in two thirds of the cases, with 64 complete penetrance at age 60 years [8]”

Liu, Z., Shan, H., Huang, J., Li, N., Hou, C., & Pu, J. (2016). A novel lamin A/C gene missense mutation (445 V > E) in immunoglobulin-like fold associated with left ventricular non-compaction. Europace, 18(4), 617-622. doi:10.1093/europace/euv044

Wada, K., Misaka, T., Yokokawa, T., Kimishima, Y., Kaneshiro, T., Oikawa, M., . . . Takeishi, Y. (2021). Blood-Based Epigenetic Markers of FKBP5 Gene Methylation in Patients With Dilated Cardiomyopathy. J Am Heart Assoc, 10(21), e021101. doi:10.1161/jaha.121.021101